# Controlling the Switch from Neurogenesis to Pluripotency during Marmoset Monkey Somatic Cell Reprogramming with Self-Replicating mRNAs and Small Molecules

**DOI:** 10.3390/cells9112422

**Published:** 2020-11-05

**Authors:** Stoyan Petkov, Ralf Dressel, Ignacio Rodriguez-Polo, Rüdiger Behr

**Affiliations:** 1Platform Degenerative Diseases, German Primate Center, GmbH, Leibniz Institute for Primate Research, 37077 Göttingen, Germany; IRodriguezPolo@dpz.eu; 2German Center for Cardiovascular Research (DZHK), partner site Göttingen, 37077 Göttingen, Germany; rdresse@gwdg.de; 3Institute for Cellular and Molecular Immunology, University Medical Center Göttingen, 37077 Göttingen, Germany

**Keywords:** iPSCs, marmoset, nonhuman primate, reprogramming, VEE-mRNA

## Abstract

Induced pluripotent stem cells (iPSCs) hold enormous potential for the development of cell-based therapies; however, the safety and efficacy of potential iPSC-based treatments need to be verified in relevant animal disease models before their application in the clinic. Here, we report the derivation of iPSCs from common marmoset monkeys (*Callithrix jacchus*) using self-replicating mRNA vectors based on the Venezuelan equine encephalitis virus (VEE-mRNAs). By transfection of marmoset fibroblasts with VEE-mRNAs carrying the human *OCT4, KLF4, SOX2*, and *c-MYC* and culture in the presence of small molecule inhibitors CHIR99021 and SB431542, we first established intermediate primary colonies with neural progenitor-like properties. In the second reprogramming step, we converted these colonies into transgene-free pluripotent stem cells by further culturing them with customized marmoset iPSC medium in feeder-free conditions. Our experiments revealed a novel paradigm for flexible reprogramming of somatic cells, where primary colonies obtained by a single VEE-mRNA transfection can be directed either toward the neural lineage or further reprogrammed to pluripotency. These results (1) will further enhance the role of the common marmoset as animal disease model for preclinical testing of iPSC-based therapies and (2) establish an in vitro system to experimentally address developmental signal transduction pathways in primates.

## 1. Introduction

Despite significant advances in modern medicine, many diseases that increase the mortality or cause disabilities in the world’s human population currently remain incurable. Due to their properties that allow them to be expanded indefinitely in vitro and to be differentiated into various somatic cell types, pluripotent stem cells (PSCs) hold significant potential for the development of cell-based therapies for a wide range of conditions, such as various neurodegenerative disorders, cancers, cardiac, and autoimmune diseases [1,2,3,4]. Moreover, the reprogramming of somatic cells to induced pluripotent stem cells (iPSCs) [5,6] has revealed an enormous potential for the development of personalized cell therapies, as autologous stem cells may minimize immunological incompatibilities while circumventing the ethical dilemmas associated with using embryo-derived stem cells (ESCs).

Currently, a number of hurdles still hinder the development of safe and efficient therapies using PSCs [7]. Despite the still unsolved issues, numerous cases of unproven and untested stem cell therapies involving adult stem cells as well as PSCs, frequently with harmful outcomes for the patients, have been reported [8]. A significant problem concerning the safety of PSC applications is teratoma formation from the transplanted cells, and various strategies have been tested to address this issue [9,10]. In this regard, the use of animal models to clarify potential hazards of PSC-based therapies in whole organisms is necessary. The common marmoset (*Callithrix jacchus*), a New World nonhuman primate (NHP), offers numerous advantages due to easiness of handling, absence of known zoonoses, high fertility, relatively short generation interval, and many close physiological similarities to humans. The development of age-related phenotypes similar to humans [11] makes the marmoset a suitable translational model for age-related conditions, such as Alzheimer [12] and Parkinson diseases [13]. Gross morphometry heart characterization also identified the common marmoset as potential model for cardiac conditions [14]. Development of efficient methods for genetic modifications in this species [15] has recently enabled the creation of several disease models for immunodeficiency, Parkinson, and polyglutamine diseases [16,17,18].

Successful generation of marmoset iPSCs has been reported by a few research groups using retroviral [19,20] or transposon [21] vectors. One disadvantage of these methods is the integration of the reprogramming transcription factors into the cell’s genome, which may cause genetic alterations. Reprogramming of marmoset cells to iPSCs has also been achieved with non-integrative episomal vectors [22]. Most recently, two research groups reported the derivation of marmoset iPSCs with synthetic mRNAs [23,24]. Since mRNAs have a short lifespan in the cells, multiple sequential transfections were necessary for successful reprogramming, subjecting the cells to increased stress. The reprogrammed iPSCs were maintained in an undifferentiated state with mouse embryonic fibroblasts (MEF)-conditioned medium [24] or on MEF feeders [23], underlining the need to optimize the xeno-free culture conditions for marmoset iPSCs, which is a prerequisite for the use of any therapeutic stem cells in preclinical trials.

In an alternative approach, the reprogramming transcription factors can be expressed by using self-replicating mRNA vectors based on the Venezuelan equine encephalitis virus (VEE-mRNAs). The 5′-capped, single-stranded viral mRNA contains four nonstructural proteins followed by the structural viral proteins, which are transcribed from an internal S26 promoter and can be replaced with heterologous proteins of choice [25,26]. A significant advantage of this vector system is that a single transfection is sufficient for the long-term transgene expression in cells of different species. By construction of VEE-mRNA expression vector containing the human *OCT4, KLF4, SOX2*, and *c-MYC* (VEE-OKS-iM), one research group produced transgene-free human iPSCs [27]. To prevent mRNA elimination by the innate immune response, B18R recombinant protein (which binds and inhibits type I interferons) was used to maintain transgene expression until the completion of reprogramming.

In this report, we describe the generation of iPSCs from marmoset somatic cells by using self-replicating Tomato-modified VEE-OKS-iM mRNA. By selective inhibition with small molecule inhibitors, we first reprogrammed marmoset fibroblasts transfected with modified VEE-OKS-iM mRNAs to intermediate primary colonies with some neural progenitor characteristics, which we then converted to pluripotent stem cells by further culturing them in customized marmoset iPSC medium. These newly generated pluripotent cells were transgene-free and were maintained long term in feeder- and serum-free conditions. At the same time, they possessed typical iPSC characteristics, such as expression of various pluripotency markers and the ability to differentiate into derivatives of the three primary germ layers in vitro and in vivo. Our study revealed a novel paradigm for flexible reprogramming of marmoset somatic cells that can be used to produce either cells of the neural stem cell lineage or iPSCs using the same primary colony population generated by a single transfection with VEE-mRNAs.

## 2. Materials and Methods

### 2.1. Isolation of Marmoset Fetal Fibroblasts (cjFFs)

Animal care as well as all treatment procedures were in accordance with the current regulations as outlined in the animal protection law and reflected in the institutional guidelines. Marmoset cjFFs were isolated from leftover fragments of marmoset fetuses from days 70–74 of gestation, first used in other unrelated projects that were fully approved by the Lower Saxony’s State Office of Consumer Protection and Food Safety (LAVES) (license numbers 42502-04-16/2129 and 42502-04-16/2130) including a positive ethics evaluation and approval by the institutional review committee. Tissues of the fetal dorsal body wall and limb buds were finely minced with scalpel blade and incubated in a mixture of 1:1 (*v*:*v*) StemPro Accutase (ThermoFisher, Darmstadt, Germany) and Collagenase IV (Worthington Biochemical Corporation, Lakewood, NJ, USA) (2 mg/mL) at 37 °C for 15 min, followed by trituration to disaggregate the tissues to single cells or small clumps. The suspension was centrifuged at 300× *g* for 5 min, resuspended in M20 culture medium (Dulbecco’s modified Eagle’s medium (DMEM) containing GlutaMAX (ThermoFisher, Darmstadt, Germany) supplemented with 20% fetal bovine serum (FBS) (ThermoFisher, Darmstadt, Germany), non-essential amino acids (NEAA (ThermoFisher, Darmstadt, Germany), penicillin/streptomycin (ThermoFisher, Darmstadt, Germany), and 5 ng/mL human basic fibroblast growth factor (bFGF) (PeproTech, Hamburg, Germany), and plated on gelatin-coated 10-cm tissue culture dishes. The cells were subsequently split 1:4–1:6 with StemPro Accutase every 3–4 days and maintained in M20 medium.

### 2.2. Synthesis of Self-Replicating mRNAs (VEE-OKS-iM-iTomato)

The plasmid T7-VEE-OKS-iM was a gift from Dr. Steven Dowdy (Addgene plasmid # 58972; http://n2t.net/addgene:58972; RRID:Addgene_58972) and was previously used by his research group for the generation of human iPSC [27]. Using conventional molecular cloning techniques, an internal ribosome entry site (IRES)-Tomato dsDNA fragment (iTomato) was inserted into the *NotI* restriction site between the stop codon of c-MYC and the IRES site of the puromycin resistance gene. The resulting plasmid (T7-VEE-OKS-iM-iTomato) was linearized by MluI restriction enzyme digest and mRNA was in vitro transcribed, capped, and polyadenylated using HiScribe ARCA T7 mRNA Kit (with tailing) (New England Biolabs, Ipswich, MA, USA) according to the manufacturer’s instructions.

### 2.3. Production of B18R-Conditioned Medium (B18R-CM)

The B18R-6His construct was excised from plasmid pTNT-B18R-6His (also a gift from Dr. Steven Dowdy (Addgene plasmid # 58979; http://n2t.net/addgene:58979; RRID:Addgene_58979)) and ligated into a *PiggyBac* transposon plasmid pTT-PB-Puro^r^, [21] by using standard molecular biology techniques, to generate pTT-PB-B18R-6His-Puro^r^ for genomic integration and constitutive protein expression. For production of B18R-CM, 2 × 10^6^ mitotically active mouse embryonic fibroblasts (MEFs) were co-transfected with 2 µg pTT-PB-B18R-6His-Puro^r^ and 1 µg pcA3-PBase-Tomato using Lipofectamine 3000 per manufacturer’s instructions. Following selection with 1 µg/mL puromycin (Sigma-Aldrich, Munich, Germany) for two weeks, the B18R-transgenic MEFs were split at 1 × 10^6^ cells/dish in 10-cm culture dishes and grown with M10 medium until reaching 100% confluency and then for another two days following change of medium. The first and second conditioned media were collected, mixed, centrifuged at 4000× *g* for 10 min to pellet cells and debris, filtered through 0.4-µm filters, and stored at −80 °C until use. The B18R-CM was added at 20% (*v*:*v*) to the cjFF culture medium 2–3 days before transfection with mRNAs as well as to the reprogramming medium thereafter.

### 2.4. Electroporation of cjFFs with VEE-OKS-iM-iTomato and Generation of Intermediate Primary Colonies

The cjFFs were transfected at P4-6 with Multiporator (Eppendorf, Hamburg, Germany) according to the protocol provided by the manufacturer. Briefly, cjFFs were suspended in hypoosmolar electroporation buffer (Eppendorf) at a concentration of 2 × 10^6^ cells/mL and incubated at room temperature for 20 min. VEE-OKS-iM-iTomato mRNA was added at 6 µg/380 µL suspension, the cells were transferred to electroporation cuvette with 2 mm gap (Sigma-Aldrich, Munich, Germany) and electroporated at 550 V, 100 µs, 1 square pulse. Following 10-min recovery at room temperature, the cells were transferred to three gelatin-coated wells (6-well plate format) and cultured at 37 °C with M20 medium supplemented with 20% B18R-CM for 9–10 days. To select the cells expressing the reprogramming mRNA, 1 µg/mL puromycin (Sigma-Aldrich, Munich, Germany) was added starting at 48–72 h post-transfection (p.t.) and was maintained until appearance of the intermediate primary colonies. After 7–8 days the selected cells were seeded on Geltrex-coated 6-cm tissue culture dishes (Starlab International, Hamburg, Germany) at 20–25 × 10^5^ cells/dish and cultured in primary reprogramming medium (StemMACS iPS-Brew XF medium (iPS-Brew; Miltenyi Biotech, Bergisch Gladbach, Germany) supplemented with 3 µM CHIR99021 (Sigma-Aldrich, Munich, Germany), 10 µM SB431542 (Selleckchem, Houston, TX, USA), and 20% B18R-CM until the picking of the intermediate primary colonies (27–30 days p.t.).

### 2.5. Generation and Culture of Marmoset iPSCs

Intermediate primary colonies with compact morphology were manually picked, digested in 1 mg/mL collagenase type IV solution (Worthington Biochemical Corporation, Lakewood, NJ, USA), and disaggregated by pipetting to single cells or small clumps. The cell suspension was seeded in 6 cm Geltrex dishes (30–40 × 10^5^ cells/dish) in marmoset iPSC medium (iPS-Brew supplemented with 3 µM IWR1 (Sigma-Aldrich, Munich, Germany), 0.5 µM CHIR99021, 0.7 µM CGP77675 (Selleckchem, Houston, TX, USA), 10 ng/mL human recombinant leukemia inhibitory factor (hrLIF) (PeproTech, Hamburg, Germany), and 7 µM Forskolin (Selleckchem, Houston, TX, USA)) and cultured for 2–3 passages in hypoxic conditions (5% O_2_, 5% CO_2_, and 90% N_2_) until the appearance of colonies with iPSC-like morphology. These colonies were manually picked, disaggregated in collagenase solution to small clumps, and passed to fresh Geltrex dishes in iPSC medium with CGP77675 concentration reduced to 0.3 µM. The iPSCs were further maintained by splitting with collagenase at ratio 1:3–1:4 every 3–4 days. To increase survival, 5 µM Pro-survival Compound (MerckMillipore, Darmstadt, Germany) was added only on the day of splitting. Hypoxic conditions were maintained during the entire duration of iPSC culture.

### 2.6. Directing Intermediate Primary Colonies into the Neural Lineages

Primary colonies were picked manually, broken to fragments (50–100 µm), and plated on Geltrex-coated dishes in neural stem cell (NSC) medium consisting of neurobasal medium (DMEM/F12 (ThermoFisher, Darmstadt, Germany) supplemented with L-glutamine, 1% N2, 2% B27, and 50 µg/mL L-ascorbic acid (Sigma-Aldrich, Munich, Germany)) supplemented with 20 ng/mL bFGF (PeproTech, Hamburg, Germany) and 20 ng/mL epidermal growth factor (EGF),(PeproTech, Hamburg, Germany). The neural rosette-like areas were manually picked, disaggregated in StemPro Accutase, and cultured further in the same conditions. For neural differentiation, NSC-like cells at 100% confluency were scraped with a 100-µL pipette tip and the cell clumps were cultured in suspension in 60-mm Petri dishes in neurobasal medium without bFGF and EGF for seven days. The resulting neurospheres were cultured on poly-L-ornithine and laminin-coated dishes for seven days and then split with StemPro Accutase on glass coverslips for immunofluorescence. For oligodendrocyte differentiation, the primary colonies were fragmented as described above and cultured in suspension in neuralizing medium (neurobasal supplemented with 1 µM Dorsomorphin (Selleckchem, Houston, TX, USA) + 2 µM SB431542 (Selleckchem, Houston, TX, USA) for 5 days. The cells were ventralized by addition of 1 µM retinoic acid (RA) (Sigma-Aldrich, Munich, Germany) and 2 µM Purmorphamine (Stem Cell Technologies, Vancouver, SC, Canada) for a further five days. The spheres were then expanded for 10 days in neurobasal medium supplemented with 1 µM smoothened agonist (SAG) (PeproTech, Hamburg, Germany) and glialized by 14 days’ culture in glial differentiation medium (GDM), consisting of neurobasal medium supplemented with 60 ng/mL T3 (Sigma-Aldrich, Munich, Germany), 100 ng/mL Biotin (Sigma-Aldrich, Munich, Germany), 10 ng/mL platelet-derived growth factor-AA (PDGF-AA) (PeproTech, Hamburg, Germany), 10 ng/mL insulin-like growth factor (IGF1) (PeproTech, Hamburg, Germany), 10 ng/mL Neurotropin 3 (NT3) (PeproTech, Hamburg, Germany), and 1 µM SAG. The glial progenitors were transferred to dishes coated with poly-L-ornithine and laminin and final differentiation was achieved by 30–40 days’ culture in GDM where PDGF-AA, IGF1, and NT3 concentrations were reduced to 5 ng/mL. Astrocyte differentiation of the intermediate primary colonies was performed by culture of the NSC-like cells generated as described above with astrocyte differentiation medium (DMEM/F12 supplemented with GlutaMAX, 1× N2, 1% FBS, 20 ng/mL Heregulin-beta 1 (PeproTech, Hamburg, Germany), 10 ng/mL bFGF, 10 ng/mL IGF1, and 10 ng/mL hrLIF) for 45–50 days.

### 2.7. Total RNA Isolation, PCR, RT-PCR, and Real-Time Relative Quantitation

Total RNA was purified using Nucleospin RNA Plus kit (Macherey-Nagel, Dueren, Germany) according to the manufacturer’s protocol. The extracted RNA was treated with DNAseI (New England Biolabs, Ipswich, MA, USA) and reverse transcription was performed with Omniscript RT Kit (Qiagen, Hilden, Germany). The PCR amplifications were conducted using Taq DNA Polymerase with Standard Taq buffer (New England Biolabs, Ipswich, MA, USA). The nucleotide sequences for RT-PCR and real-time PCR primers (synthesized by Sigma-Aldrich, Munich, Germany) are shown in Table 1. Relative quantitation real-time qPCR was performed with StepOnePlus Real-time PCR system (Applied Biosystems, Bedford, MA, USA) using Power SYBR Green PCR master mix (Applied Biosystems, Bedford, MA, USA). Three independent biological samples from each experimental group (cjFFs, Primary, and iPSCs) with at least two technical replicates/sample were included in each analysis. Statistical analysis (one-way ANOVA with Tukey HSD (honestly significant difference) test) and graphical presentation of the results were performed with R-studio.

### 2.8. Immunofluorescence and Alkaline Phosphatase Activity

All procedures were performed at room temperature. The iPSCs intended for immunofluorescence analysis were cultured on Geltrex-coated glass coverslips. The cells were fixed in 4% paraformaldehyde solution for 15 min, permeabilized with 0.15% Triton X for 10 min, and blocked with 1% BSA for 1 h prior to incubation with primary antibodies raised against OCT4 (Cell Signaling Technology, Danvers, MA, USA; Cat. # 28901:800), NANOG (Cell Signaling Technology, Danvers, MA, USA; Cat. # 4903; 1:500), SOX2 (Cell Signaling Technology, Danvers, MA, USA; Cat. # 3728; 1:200), SSEA-4 (MerckMillipore, Darmstadt, Germany; Cat. # MAB4304; 1:100), TRA-1-60 (eBioscience, San Diego, CA, USA; Cat. # 14-8863; 1:100), TRA-1-81 (eBioscience, San Diego, CA, USA; Cat. # 14-8883; 1:100), SALL4 (Sigma-Aldrich, Munich, Germany; Cat. # HPA015791; 1:200), CDH1 (Cell Signaling Technology, Danvers, MA, USA, Cat. # 3195; 1:400), or OTX2 (Sigma-Aldrich, Munich, Germany; Cat. # HPA000633; 1:200). Differentiated neuronal cells were processed as described above and immunostained with anti-β-III-Tubulin (Sigma-Aldrich, Munich, Germany; Cat. # T8660; 1:100), anti-NESTIN (MerckMillipore, Darmstadt, Germany; Cat. # MAB5326; 1:100), or anti-microtubule-associated protein 2 (MAP2) (Sigma-Aldrich, Munich, Germany; Cat. # HPA012828; 1:100). Oligodendrocytes and astrocytes were identified with anti-α-Tubulin (DM1A; ThermoFisher, Darmstadt, Germany; Cat. # 62204; 1:200), anti-O4 (R&D, Minneapolis, MN, USA; Cat. # MAB1326; 5µg/mL), or anti-glial fibrillary acidic protein (GFAP) (DAKO, Glostrup, Denmark; Cat. # Z0334; 1:500). Endodermal cells were visualized with anti-SOX17 (ThermoFisher, Darmstadt, Germany; Cat. # PA5-23352; 1:200) or anti-alpha Fetoprotein (AFP) (DAKO, Glostrup, Denmark; Cat. # A0008; 1:300). Cardiomyocytes were immunostained with anti-α-Actinin (Sigma-Aldrich, Munich, Germany; Cat. # A7811; 1:100), anti-cTnT (ThermoFisher, Darmstadt, Germany; Cat. # MS-295-PABX; 1:200), anti-MLC2a (Synaptic Systems, Goettingen, Germany; Cat. # 311-011; 1:200), anti-Titin (MerckMillipore, Darmstadt, Germany; Cat. # MAB1553; 1:50), and anti-CX43 (Abcam, Cambridge, United Kingdom; Cat. # ab11370; 1:1000). Visualization was performed with Alexa Fluor 488 donkey anti-rabbit, Alexa Fluor 488 goat anti-mouse, or Alexa Fluor 594 donkey anti-mouse secondary antibody (Invitrogen, Carlsbad, CA, USA) (all diluted 1:1000). Alkaline phosphatase activity was determined using Alkaline Phosphatase kit (Sigma-Aldrich, Munich, Germany) as instructed by the manufacturer.

### 2.9. In Vitro Differentiation of Marmoset iPSCs

Embryoid bodies were generated by treating iPSC cultures with Versene (ThermoFisher, Darmstadt, Germany) for 3 min at room temperature, scraping off, and culturing the cell clumps in Petri dishes in Iscove’s medium (IMDM with GlutaMAX (ThermoFisher, Darmstadt, Germany) supplemented with 20% FBS, NEAA, and 450 µM 1-monothyoglycerol (Sigma-Aldrich, Munich, Germany)) for 7–10 days. The EBs were then cultured on Geltrex-coated glass coverslips for another week and processed for immunofluorescence. Directed cardiomyocyte differentiation was performed with iPSCs at P9-12 as described by [28]. Neural differentiation was achieved by neuralization in DMEM/F12 supplemented with 10% Knockout Serum Replacement (ThermoFisher, Darmstadt, Germany), NEAA, 50 µg/mL ascorbic acid (Sigma-Aldrich, Munich, Germany), 2 μM SB431542 (Selleckchem, Houston, TX, USA), and 1.5 μM dorsomorphin (Selleckchem, Houston, TX, USA) for 5–7 days. Then, the cells were split with StemPro Accutase and cultured in NSC medium. To differentiate into neurons, the cells were cultured on poly-L-ornithine and laminin-coated dishes or glass coverslips in neurobasal medium for 10–14 days. For differentiation into endoderm, the iPSCs were first cultured in basal medium (RPMI-1640 with 2% B27) supplemented with 100 ng/mL Activin A (PeproTech, Hamburg, Germany) and 3 µM CHIR99021 for three days and then with 5 ng/mL bFGF, 20 ng/mL BMP4 (PeproTech, Hamburg, Germany), and 0.5% DMSO (Sigma-Aldrich, Munich, Germany) for five more days. At the end of experiment the differentiated cells were passed on Geltrex-coated glass coverslips with StemPro Accutase and immunostained with anti-AFP antibodies.

### 2.10. Teratoma Assay

Marmoset iPSCs at P20-30 were harvested with collagenase and resuspended in cold Geltrex solution prepared with iPSC culture medium at concentration of 1 × 10^7^ cells/mL together with 1 × 10^5^ cells/mL mitotically inactivated mouse fetal fibroblasts (MEFs). The cells were transported on ice to the mouse housing facility where they were injected subcutaneously into the left flank of immunodeficient SCID/beige mice (C.B-17/IcrHsd-scid-bg) at 100 µL/injection/mouse. The mice were bred in the central facility for animal experimentation at the University Medical Center Göttingen under specific pathogen-free conditions in individually ventilated cages and in a 12-h light–dark cycle. The mouse experiments were approved by the local government (33.9-42502-04-19/3074) and were carried out in compliance with EU legislation (Directive 2010/63/EU). Mice with tumor growth (detected by regular palpation) were sacrificed before the tumors reached 1 cm size approximately 1–3 months after injection. All remaining animals were sacrificed after three months and autopsies were performed to exclude an unrecognized tumor growth.

The paraffin embedding, sectioning, and immunohistochemical staining of the teratomas were performed as described previously [21]. The presence of lineage-specific markers was determined by incubation with anti-β-III-Tubulin, anti-NESTIN, anti-PAX6 (MerckMillipore, Darmstadt, Germany; Cat. # AB2237; 1:500), anti-smooth muscle actin (SMA) (Sigma-Aldrich, Munich, Germany; Cat. # A2547, 1:1000), anti-SOX17, and anti-SOX9 (MerckMillipore, Darmstadt, Germany; AB5535, 1:300) antibodies and visualization with EnVision FLEX Mini kit (DAKO, Glostrup, Denmark).

### 2.11. Karyotyping

The chromosomal numbers of four iPSC lines were determined at P21-50 as described earlier [29]. Briefly, the cultures were arrested at metaphase by adding 0.02 mg/mL Demecolcine (Sigma) to the culture medium for 1.5 h and then disaggregated with StemPro Accutase to single cells. Following incubation in hypoosmotic 0.56% KCl solution for 20 min, the cells were fixed in 3:1 methanol/acetic acid (*v*:*v*) fixative and metaphase spreads were generated by dripping 50 µL drops of the cell suspension on glass slides positioned at a slight angle over a steaming water bath. The metaphase spreads were air-dried, stained in Giemsa solution (Sigma) for 10 min, and photographed with Axiophot microscope (Zeiss, Oberkochen, Germany) equipped with CRI Nuance multispectral imaging camera. Approximately 25–30 images per sample were used for chromosome counting.

## 3. Results

### 3.1. Derivation of Primary Colonies and iPSCs

The configuration of the modified T7-VEE-OKS-iM-iTomato plasmid and an outline of the reprogramming process are shown in Figure 1A,B. Successfully transfected marmoset fetal fibroblasts (cjFFs) were recognizable by their Tomato fluorescence (Tomato^+^) (Figure 1C). Following selection with Puromycin for 5–7 days, nearly all cells were Tomato^+^. Initially, we failed to obtain any iPSC lines when we cultured these cells on Geltrex-coated dishes with iPS-Brew or E8 medium, which are media designed for the feeder-free culture of human pluripotent stem cells. This prompted us to test different small molecule inhibitor supplements for marmoset cell reprogramming. In one of these experiments, primary colonies with compact morphology and clearly defined borders appeared in iPS-Brew supplemented with CHIR99021 and SB431542 (Figure 1D,E). These colonies were Tomato^+^ (Figure 1D) and varied widely in size between 200 and 3000 µm across. The primary colonies also exhibited alkaline phosphatase (AP) activity (Figure 1F). We manually picked 10–15 colonies/dish, disaggregated them in collagenase, and cultured them as single cells or small clumps in customized marmoset iPSC medium (iPS-Brew supplemented with IWR1, CHIR99021, CGP77675, rhLIF, and Forskolin). After 2–3 passages, we observed colonies with morphology characteristic of primate iPSCs (Figure 1G). These iPSC-like colonies were picked manually and further expanded and maintained by splitting with collagenase type IV (Figure 1H). In total, four iPSC lines (three male and one female) were established from three different primary cjFF samples and were maintained for 50–72 passages by the time of writing this report.

### 3.2. Gene Expression Analysis

All iPSC colonies possessed strong AP activity (Appendix AA) and were shown to be positive for expression of OCT4A, NANOG, SSEA-4, TRA-1-60, TRA-1-81, CDH1, SALL4, and SOX2 by immunofluorescence (Figure 2A). Endogenous transcripts of *OCT4A* and *DPPA2* were detected by RT-PCR only in iPSCs, which clearly distinguished them from primary colonies and cjFFs (Figure 2B). Other pluripotency-related genes, such as *NANOG, GDF3, TERT, ZFP42, DPPA4,* and *CDH1,* were detected at mRNA level in iPSCs as well as in primary colonies (Figure 2B). The expression of the exogenous reprogramming factors in intermediate primary colonies was confirmed at mRNA level by RT-PCR (Figure 2C). In absence of B18R-CM, all putative iPSC colonies were Tomato^-^ by P6, and the transgenes were not detected in iPSCs at mRNA or DNA level (Figure 2C). Although most pluripotency-related mRNAs were detected in primary colonies as well as iPSCs by conventional RT-PCR, real-time relative quantitation RT-PCR analysis revealed that *OCT4A, NANOG, GDF3, CDH1, LIN28,* and *DPPA4* were significantly upregulated in iPSCs relative to cjFFs and primary colonies (Figure 2D,E). Only *SALL4* expression in primary colonies was already in the range of *SALL4* expression in iPSCs (Figure 2E).

Neural markers *NESTIN*, *NOTCH1*, and *GBX2* as well as neural plate border markers *PAX3* and *MSX1* were upregulated in intermediate primary colonies relative to cjFFs (Figure 3A–C). Following the second reprogramming step, the neural markers *NESTIN*, *NOTCH1*, *CDH2*, *PAX3*, *MSX1*, and *DLX5* became significantly downregulated in the iPSCs (Figure 3B,C). At the same time, the iPSCs significantly upregulated *SOX2* and *OTX2* relative to both cjFFs and primary colonies (*p*-values < 0.001 and < 0.03, respectively) (Figure 3A). The expression of OTX2 in iPSCs was also confirmed by immunofluorescence (Appendix A). Finally, genes associated with endo- or mesodermal differentiation, *GATA6, MIXL1*, and *NKX2.5,* were significantly downregulated in primary colonies and iPSCs relative to cjFFs (Figure 3D) (all *p*-values ≤ 0.01). Other differentiation genes, such as *HAND1*, *GATA4*, and *TBX5,* also showed downregulation in primary colonies and iPSCs (Figure 3D); however, it was not found statistically significant. Altogether, the analysis of the pluripotency-related and lineage-specific markers allows an accurate discrimination between the fetal fibroblasts, the primary intermediate colonies, and the putative iPSCs. Furthermore, the neural marker analysis indicates an ectodermal/neuronal identity of the primary intermediate colonies.

### 3.3. In Vitro and In Vivo Differentiation of Marmoset iPSCs

When cultured in suspension, the iPSCs formed EBs, some of which became cavitated within seven days (Appendix A). A small number (2–3%) of the EBs also showed contracting activity due to presence of immature cardiomyocytes (Appendix A). On Geltrex-coated glass coverslips, the EBs formed outgrowths containing β-III-Tubulin-positive neurons (Appendix A) and SOX17-positive endoderm-like cells (Appendix A). We performed directed neural differentiation by first “neuralizing” the iPSCs with dual SMAD inhibition (dorsomorphin and SB431542) followed by splitting and culture in NSC medium, where the cells maintained expression of NSC markers PAX6 and Nestin (Appendix A). When these NSC-like cells were cultured in neurobasal medium on poly-ornithine and laminin, they formed neural rosettes (Appendix A) and subsequently differentiated into neurons (Appendix A) positive for β-III-Tubulin and NESTIN (Figure 4A,B). Directed differentiation of the iPSCs into endoderm resulted in appearance of typical “cobblestone”-like cell layer (Figure 4C), where approximately 40% of the cells stained positive for AFP at different intensities by immunofluorescence (Figure 4D). Finally, directed cardiac differentiation resulted in contracting myocytes organized in several clusters (Appendix A). These primitive cardiomyocytes were metabolically selected, transferred onto glass coverslips, and stained for expression of cardiac-specific proteins. The majority of the cardiomyocytes was positive for cTnT, MLC2a, α-actinin, CX43, and Titin (Figure 4E–H).

When injected into immunodeficient mice, the iPSCs formed teratomas with presence of primitive gut-like endoderm (Figure 4I), gland-like structures (Figure 4J), and bone tissues (Figure 4K,L). The endodermal cells were positive for SOX9 (Figure 4M), while presence of mesoderm was confirmed by SMA staining (Figure 4N). Ectodermal differentiation was demonstrated by the presence of β-III-Tubulin, PAX6, and NESTIN-positive cells (Figure 4O–Q). In summary, these data demonstrate pluripotency of the iPSCs.

### 3.4. Karyotyping

Three iPSC lines were processed for karyotyping and at least 25 images/line, where all chromosomes were clearly distinguishable, were used for counting. All lines had normal chromosome numbers where two were male (46 XY) and one was female (46 XX) (Appendix A).

### 3.5. Neurogenic Potential of the Intermediate Primary Colonies

When the intermediate primary colonies were mechanically picked, broken to small fragments, and cultured in NSC medium, multiple neural rosette-like structures formed in areas with high cell density (Figure 5A). These rosettes were picked, disaggregated, and cultured as monolayer in NSC medium (Figure 5B). For the entire period of culture (7–10 passages) the rosette-derived cells maintained SOX2 and NESTIN expression (Figure 5C). Differentiation was conducted by first generating neurosphere-like aggregates in suspension for seven days (Figure 5D) and then cultured on poly-ornithine and laminin, whereupon the cells differentiated into neurons (Figure 5E) expressing β-III-Tubulin and MAP2 (Figure 5F). In addition, the primary colony cells differentiated into GFAP-positive astrocytes (Figure 5G,H) or oligodendrocytes expressing alpha-tubulin (Figure 5I) and O4 (Figure 5J). We concluded from these data that the intermediate colonies have neurogenic potential when cultured under appropriate conditions.

In order to find out whether the developmental potential of the intermediate primary cells extends to the other lineages (i.e., whether the cells are pluripotent), the colonies were cultured in conditions promoting EB formation as described for iPSCs. Unlike the EBs generated from iPSCs, the clumps became darker (Appendix A), aggregated with each other, and did not expand in size. In one of the three lines tested, the cells died and the clumps disintegrated. In the remaining cultures, no cavitation or beating clusters indicative of CM formation were observed. When allowed to attach on Geltrex-coated dishes, the clump outgrowths contained multiple neuronal-like cells (Appendix A) and stained positive for nestin (Appendix A) and β-III-Tubulin (Appendix A). At the same time, we did not find any cells positive for SOX17, AFP, or SMA by immunofluorescence. Considering these results together with the gene expression data, it can be concluded that the intermediate primary cells did not possess pluripotency properties comparable with iPSCs.

Since long-term self-renewal is an important characteristic of neural stem cells, we continued to culture some primary colonies with the same culture medium used in the first reprogramming step (iPS-Brew supplemented with CHIR99021 and SB431542). The resulting cell lines retained typical for neural progenitors’ colony morphology (Appendix A) and maintained alkaline phosphatase (Appendix A) and SOX2 expression (Appendix A) for over 100 passages without showing any signs of slowing proliferation. These primary colony derivatives also maintained their ability to differentiate into neurons and glia when subjected to the already described differentiation protocols. These results indicate that the intermediate primary colonies possess the ability for self-renewal when supported by proper inhibitor supplementation.

### 3.6. Small Molecule Inhibitors Are Necessary for Maintaining Pluripotency of Marmoset iPSCs

The small molecule inhibitors IWR1, CHIR, and CGP77675 were necessary for the generation of iPSC colonies in the second step of the reprogramming, as no colonies were observed when these compounds were omitted. In order to determine the role of individual medium supplements in the maintenance of pluripotency during long-term culture, we cultured three iPSC lines for 5–6 passages with medium where either all (iPS-Brew only) or just one of the five supplements were removed (-IWR1, -CHIR99021, -CGP77675, -hrLIF, or -Forskolin). At the end of the experiment, the relative expression levels of key pluripotency markers OCT4A, NANOG, SALL4, GDF3, and DPPA4 were determined (Figure 6).

As expected, the removal of all inhibitors caused loss of iPSC-like morphology within three passages (Figure 6A), but the observable changes in the other experimental groups were more subtle. The colonies cultured without IWR1, CHIR99021, CGP77675, or Forskolin partially lost their compact morphology by P5 together with the appearance of many differentiated cells, while the removal of hrLIF had no apparent effect (Figure 6A). Real-time qPCR analysis showed that removal of IWR1 led to significantly reduced expression levels of OCT4, NANOG, and DPPA4 (Figure 6B). Similarly, cultures maintained without CGP77675 showed significant downregulation of OCT4 and NANOG. In addition, absence of CHIR99021 reduced significantly the expression of OCT4A. Although SALL4 and GDF3 also appeared to be affected by IWR1, CHIR99021, or CGP77675 withdrawal, the changes in their expression levels were not found to be statistically significant. Finally, omission of hrLIF or Forskolin from the culture medium did not significantly affect the expression of any of the five examined pluripotency markers (Figure 6B).

## 4. Discussion

Among the non-integrative methods for iPSC production, episomal plasmids have been a popular choice due to their relatively lower costs and simplicity of use. One of the few caveats of this expression system is that the episomes currently used by most groups contain immortalization factors, such as large T antigen [30] or shRNA for p53 knockdown [31], which in some cases have been shown to cause genomic instability [32,33]. In addition, episomal plasmid expression may persist in up to 33% of the reprogrammed cells even after P11 [31,32]. Using mRNAs for iPSC generation eliminates the risks of genomic integration and allows for quick and efficient removal of the transgenes. At the same time, the short half-life of the mRNA messages in the cells is a major disadvantage since a sustained expression of the reprogramming factors over time is necessary for successful reprogramming. For this reason, nonreplicating mRNAs need to be introduced multiple times into the cells. In order to mitigate the low reprogramming efficiency, additional reprogramming factors were included: various miRNAs [23] or *NANOG* and *LIN28* transgenes as well as shRNA-p53 [24]. In contrast to these reports, we were able to derive marmoset iPSCs with a single transfection of self-replicating VEE-mRNAs carrying only the four Yamanaka factors (*OCT4, KLF4, SOX2*, and *c-MYC*) and, importantly, devoid of any immortalization factors. However, it must be noted that this was possible only after modification of the protocol to include an initial reprogramming step with generation of partially reprogrammed intermediate primary colonies by culture with GSK3β and ALK5 inhibitors (CHIR99021 and SB431542, respectively) because protocols optimized for reprogramming of human cells failed to produce any marmoset iPSCs. This particular inhibitor combination was included in our experiments because previous reports have shown that CHIR99021 can induce reprogramming of mouse somatic cells transfected only with two factors (OCT4 and KLF4) [34]. Moreover, dual inhibition of the transforming factor beta (TGFβ) and mitogen-activated protein kinases (MAPK/ERK) signaling pathways with SB431542 and PD0325901, respectively, has been shown to enhance the reprogramming of human fibroblasts >200-fold [35]. By using another inhibitor with similar action to SB431642 (A-83-01) together with PD0325901, Zhu and coworkers [36] were able to reprogram human keratinocytes transfected with only one reprogramming factor (OCT4). These authors also reported that adding CHIR99021 to the inhibitor mix significantly increased the reprogramming efficiency.

Under the influence of the two small molecule inhibitors, the transfected cjFFs initiated robust proliferation and formed multiple primary colonies, which we at first mistook for iPSCs due to their compact morphology and clearly defined borders. Subsequently, we found out that these cells were negative for endogenous OCT4A and did not express the other key pluripotency markers we examined at levels comparable with iPSCs, with the sole exception of *SALL4*. Instead, they showed significant upregulation of neural progenitor markers (*SOX2*, *NESTIN*, *NOTCH1*, *CDH2*, *GBX2*, and *PAX3*). These findings prompted us to direct them into the neural lineage and we were successful in producing SOX2^+^ and NESTIN^+^ NSC-like cells that could be maintained in NSC culture conditions and were capable of neuronal differentiation. Additionally, the primary colonies demonstrated competency to enter the glial lineage by forming oligodendrocytes and astrocytes in vitro, thus affirming their neural progenitor status. On the other hand, no differentiation into any of the other primary germ layers (endoderm/mesoderm) was observed under culture conditions that had proven efficient in directing EB formation from iPSCs, suggesting that the intermediate primary cells lack pluripotency characteristics. Nevertheless, they could be expanded and maintained in medium supplemented with CHIR99021 and SB431542 for over 100 passages, which demonstrated their ability for self-renewal, an important characteristic of neural progenitor stem cells. Even after many passages, the primary colony derivatives still expressed SOX2, NESTIN, and NOTCH1, but lacked OCT4A expression and failed to form all three primary germ layers in vitro and in vivo (results not shown). A detailed description of the characterization results here would exceed the scope of the current manuscript and will be published in a separate report.

The reasons for the inability of CHIR99021 and SB431542 to support the reprogramming of marmoset fibroblasts to pluripotency are currently not clear, but they could be due to some of the differences between the experimental conditions used by us and those reported by other research groups. Besides the species origin of the cells and the expression system used, a significant deviation from most of the published protocols was the omission of PD0325901 from our reprogramming medium, since the presence of this inhibitor completely abolished the formation of primary colonies and caused reduced growth and massive differentiation when introduced into cultures of established marmoset iPSCs (results not shown). On the other hand, our results bear many similarities with a report where the same two inhibitors generated neural progenitors from human and NHP fibroblasts transduced with Sendai virus expressing the Yamanaka’s reprogramming factors [37]. Additionally, it has been confirmed that the modulation of Wnt and TGF3β signaling by the combination of CHIR99021 and SB431542 favors the establishment of neural precursors from human pluripotent cells [38,39]. While the generation of neural progenitor-like cells in the first reprogramming step corroborates findings already described in the published literature, we showed, for the first time, that these particular cells have the potential to be further reprogrammed to pluripotency. The effectiveness of this two-step reprogramming approach is in agreement with previous research, which has shown that adult NSCs can be efficiently reprogrammed to pluripotency [40], while another group generated murine iPSCs by first inducing intermediate NSC-state using Id3 and OCT4 [41]. When cultured with modified iPSC medium containing Wnt signaling inhibitor IWR1, low concentration of TGF3β inhibitor CHIR99021, and Src kinase inhibitor CGP77675, the intermediate primary colonies formed iPSC colonies within 2–3 passages. This conversion was dependent on the presence of IWR1, which has been shown to stabilize AXIN1/2 and cause cytoplasmic retention of β-catenin by preventing its translocation to the nucleus [42]. It has been demonstrated that cytoplasmic β-catenin interacts with the transcriptional co-activator with PDZ-binding motif (TAZ) to promote self-renewal of mouse epiblast stem cells as well as human iPSCs [43]. In addition, together with CHIR99021, Src kinase inhibitor CGP77675 has been shown to maintain the developmental competence of mouse ESCs [44]. We found that, similarly to IWR1, CGP77675 was important for the generation of marmoset iPSCs under feeder-free conditions as well as for their long-term maintenance, as its withdrawal from the culture medium later caused downregulation of OCT4 and NANOG and a significant change in the morphology of the iPSC colonies. Lastly, although rhLIF and Forskolin were not necessary for the derivation of marmoset iPSCs, they bolstered the reprogramming process by improving cell proliferation. The combination of IWR1, CHIR99021, and a Src inhibitor has already been used in the generation of porcine and human expanded potential stem cells [45], while in our hands, IWR1 and CHIR99021 have recently proven very helpful in the long-term maintenance of iPSCs from NHPs, namely, rhesus monkey and baboon [46]. Altogether, we showed in the present report as well as in our previous publication [46] that NHP pluripotent stem cells have cell culture requirements different from human cells. This may reflect variations in the relevance of specific signal transduction pathways for the maintenance of pluripotency in the different primate species. Under the influence of the inhibitor cocktail, many endogenous pluripotency markers became upregulated, including OCT4A, which was not detectable in cjFFs or primary colonies. At the same time, neural progenitor markers were downregulated in the iPSCs, with the exception of SOX2 and OTX2. Both markers play important roles in the development of the nervous system; however, SOX2 is also involved in pluripotency, while OTX2 is expressed in the epiblast and is characteristic of mouse epiblast stem cells and human ESCs [47]. Therefore, the relatively high expression of OTX2 in our marmoset iPSCs suggests that they are primed pluripotent cells. Lastly, GBX2 was also present in the fully reprogrammed cells, albeit insignificantly downregulated relative to primary colonies. The expression of this important neural development marker in iPSCs is not in conflict with their pluripotency status, as it has been shown that overexpression of GBX2 can enhance reprogramming of mouse fibroblasts to iPSC and to maintain mouse ESCs in undifferentiated state in absence of leukemia inhibitory factor/ signal transducer and activator of transcription 3 (LIF/STAT3 signaling [48].

The reprogramming protocol employed in this study differs from the majority of protocols described to date for the production of iPSCs from NHPs and other species. One advantage of this novel approach is the generation in the first reprogramming step of multiple primary colonies, which then can be flexibly used for the derivation of neural and/or pluripotent cells, according to the objectives of the particular study (Figure 7). Despite providing a proof of principle, it remains to be seen whether this paradigm would also function in cells of species other than the common marmoset. Nevertheless, taking into consideration the data obtained from this study, we expect that our results will help to further improve the conditions for reprogramming and culturing of marmoset iPSCs. We also expect that our new reprogramming method, resulting in transgene-free iPSCs, in combination with the fully defined culture medium, will enhance the use of the marmoset monkey as a NHP species in translational studies of regenerative medicine. Finally, the relatively efficient genetic modification of marmoset monkeys allows the testing of key findings made in stem cell culture in long-term in vitro embryo culture [49,50] or even in vivo.

## 5. Conclusions

By using self-replicating VEE-mRNAs and small molecule inhibitors, we successfully derived transgene-free marmoset iPSCs in feeder- and serum-free conditions. By inhibition of GSK3β and the TGF3β receptor signaling, we first established intermediate primary colonies with capability to differentiate into the neural lineage. Subsequently, we converted these primary colonies to iPSCs by modulating the β-catenin signaling and by inhibiting the Src kinase. With our experiments, we established a novel paradigm for flexible reprogramming of marmoset somatic cells that will enhance the role of the common marmoset as biomedical nonhuman primate model for the development of stem cell therapies.

## Figures and Tables

**Figure 1 cells-09-02422-f001:**
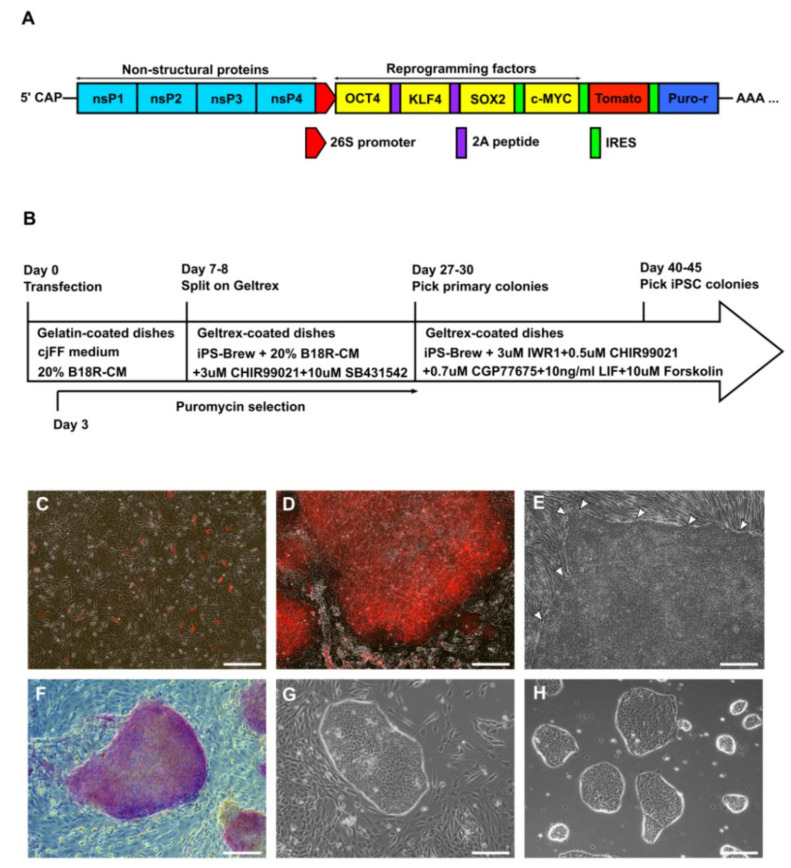
Generation of marmoset induced pluripotent stem cells (iPSCs) with the Tomato-modified self-replicating mRNA based on the Venezuelan equine encephalitis virus (VEE-OKS-iM-iTomato). (**A**) Structure of the reprogramming VEE-OKS-iM-iTomato mRNA. (**B**) Scheme of the reprogramming process. (**C**) Image of marmoset fetal fibroblasts (cjFFs) transfected with VEE-OKS-iM-iTomato at day 3 post-transfection. Transfected cells are recognizable by their red fluorescence. (**D**) An intermediate primary colony at day 27 post-transfection with red VEE-OKS-iM-iTomato fluorescence. (**E**) An intermediate primary colony at day 27 post-transfection with characteristic compact morphology and clearly defined borders (indicated with arrowheads). The cells beyond the upper and left borders of the colony are non-reprogrammed cjFFs. (**F**) An intermediate primary colony stained for alkaline phosphatase (AP). (**G**) Marmoset iPSC colony after second round of reprogramming of the intermediate primary cells with IWR1, CHIR99021, CGP77675, human recombinant leukemia inhibitory factor (hrLIF), and Forskolin. (**H**) Morphology of marmoset iPSC colonies growing on Geltrex at P27. (All scale bars = 200 μm).

**Figure 2 cells-09-02422-f002:**
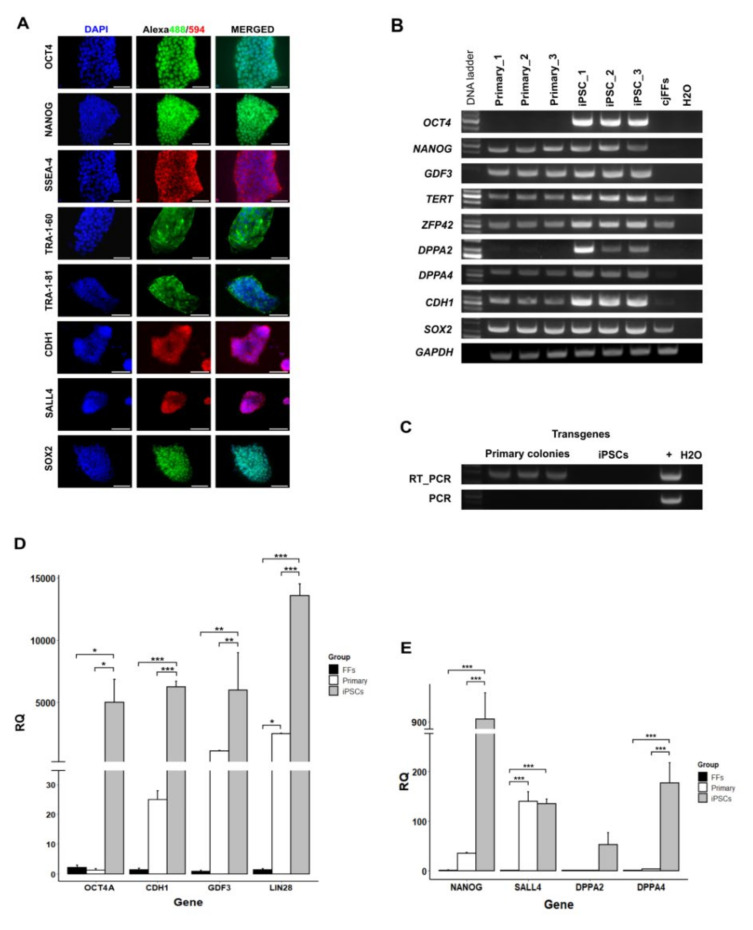
Expression of pluripotency-related genes in marmoset iPSCs. (**A**) Immunofluorescence of iPSC colonies stained for expression of OCT4, NANOG, SSEA-4, TRA-1-60, TRA-1-81, CDH1, SALL4, and SOX2. (Scale bars = 50 μm). (**B**) RT-PCR analysis of endogenous pluripotency marker expression in primary cultures and iPSCs. (**C**) RT-PCR and PCR detection of transgenes in primary colonies and iPSCs. Plasmid DNA used as positive control (denoted as “+”). (**D**) Real-time RT-qPCR analysis of relative expression levels of *OCT4A*, *CDH1*, *GDF3*, and *LIN28* in cjFFs, intermediate primary colonies, and iPSCs. (**E**) Comparison of the relative expression levels of *NANOG*, *SALL4*, *DPPA2*, and *DPPA4* in cjFFs, intermediate primary colonies, and iPSCs (each group n = 3) by real-time RT-qPCR (Relative Quantity (RQ) = 2^−ddCt^). (Data are presented as mean + standard error of the mean (SEM). Statistically significant differences between different experiment groups are indicated by asterisks as follows: * *p* < 0.05, ** *p* < 0.01, *** *p* < 0.001.).

**Figure 3 cells-09-02422-f003:**
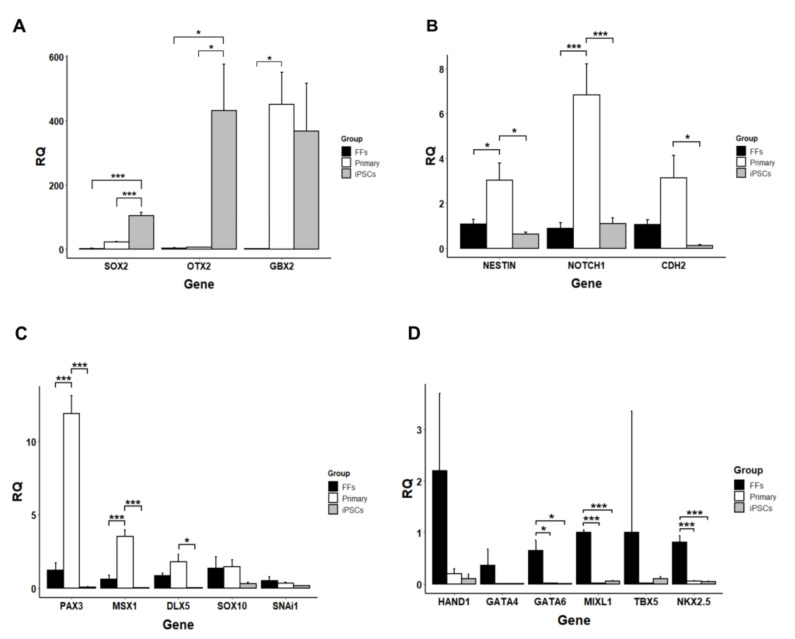
Real-time RT-qPCR analysis of lineage-specific marker expression. (**A**) Relative expression of *SOX2*, *OTX2*, and *GBX2*. (**B**) Relative expression of neural stem cell markers *NESTIN*, *NOTCH1*, and *CDH2* in cjFFs, intermediate primary colonies, and iPSCs by real-time RT-qPCR. (**C**) Relative expression of neural crest markers *PAX3, MSX1, DLX5, SOX10,* and *SNAi1*. (**D**) Relative expression of endo/mesodermal markers *HAND1*, *GATA4, GATA6, MIXL1, TBX5,* and *NKX2-5* (RQ = 2^−ddCt^). (Data are presented as mean + SEM. Each biological group n = 3. Significant differences are indicated with asterisks as follows: * *p* < 0.05, ** *p* < 0.01, *** *p* < 0.001).

**Figure 4 cells-09-02422-f004:**
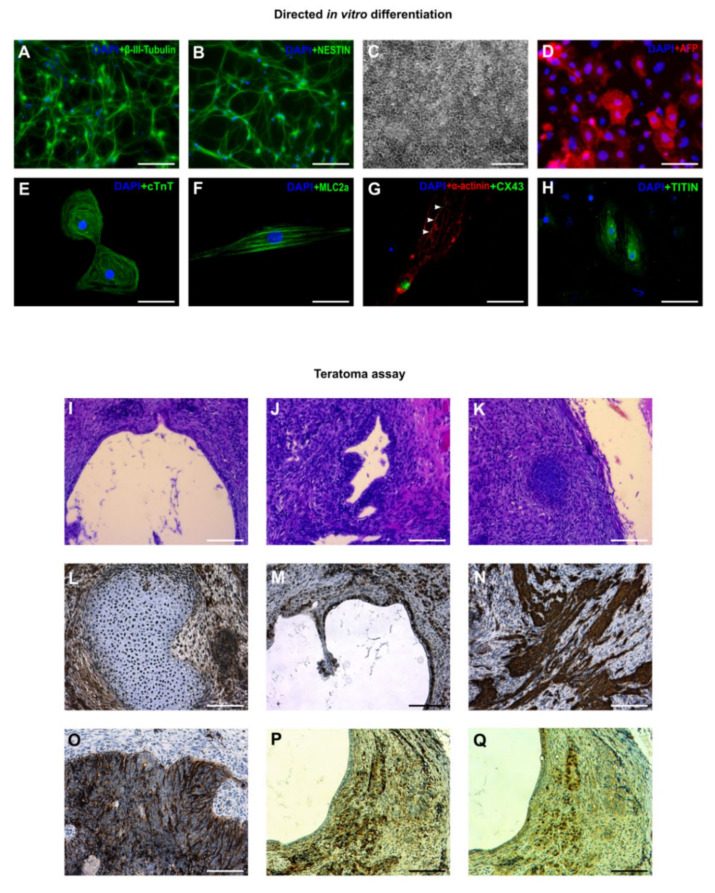
In vitro and in vivo differentiation of marmoset iPSCs. (**A**,**B**) Neurons positive for β-III-Tubulin and NESTIN. (**C**) Endodermal differentiation with cobblestone-like morphology. (**D**) Primitive endoderm stained with anti-alpha Fetoprotein (AFP). (**E**–**H)** Cardiomyocytes stained for: (**E**) cTnT, (**F**) MLC2a, (**G)** α-actinin and CX43 (arrowheads), and (**H)** Titin. (**I**–**Q**) Teratoma assay: (**I**) Gut-like endoderm. (**J**) Gland-like endoderm. (**K**) Bone. (**L**) Bone tissue stained for SOX17 (which is also involved in osteogenesis). (**M**) Gut-like tissues stained with anti-SOX9. (**N**) Smooth muscle stained for smooth muscle actin (SMA). (**O**) Neuronal-like cells stained for β-III-Tubulin. (**P**,**Q**) Successive sections stained for presence of NESTIN and PAX6. (Scale bars: (**A**,**B**,**D**–**H**) = 50 um; (**C**) = 200 μm; (**I**–**Q**) = 100 μm).

**Figure 5 cells-09-02422-f005:**
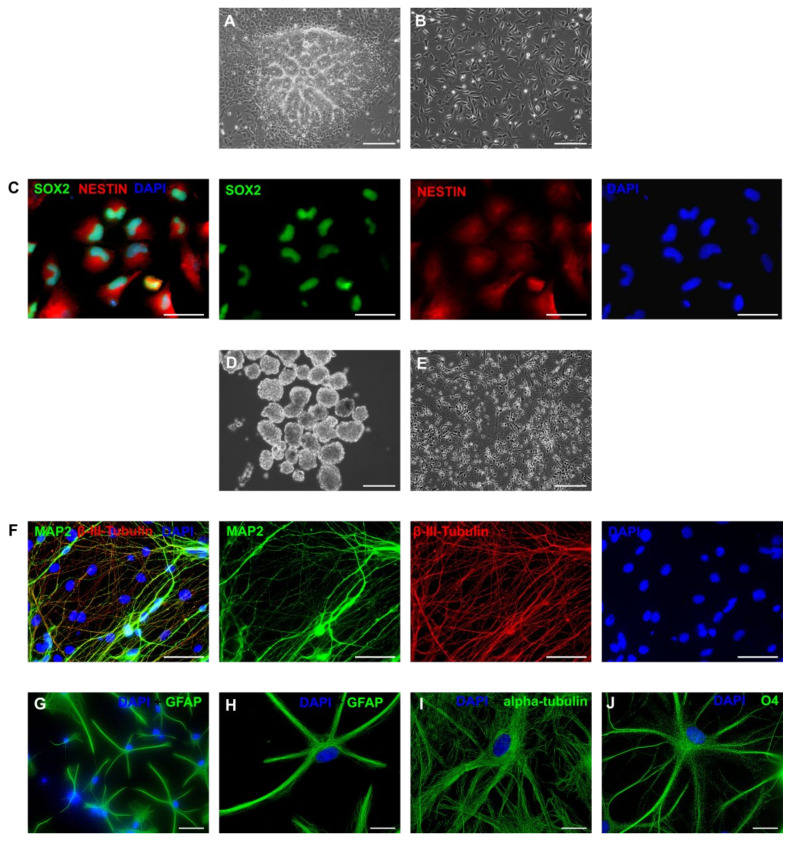
Directing the intermediate primary colonies into the neural lineage. (**A**) Neural rosettes in P1 after culture in neural stem cell (NSC) medium containing basic fibroblasts growth factor (bFGF) and epidermal growth factor (EGF). (**B**) NSC-like cells growing on Geltrex in NSC medium. (**C**) NSC-like cells expressing SOX2 and NESTIN. (**D**) Neural sphere-like aggregates in suspension culture. (**E**) Neuronal cells at the end of differentiation. (**F**) Immunofluorescence from neurons stained with anti-microtubule-associated protein 2 (MAP2) and anti-β-III-Tubulin. (**G**,**H**) Astrocytes expressing glial fibrillary acidic protein (GFAP). (**I**,**J**) Oligodendrocytes stained with anti-α-Tubulin (**I**) and anti-O4 (**J**). (Scale bars: (**A**–**D**) = 200 μm; (**E**–**G)** = 50 μm; (**H**–**J**) = 20 μm).

**Figure 6 cells-09-02422-f006:**
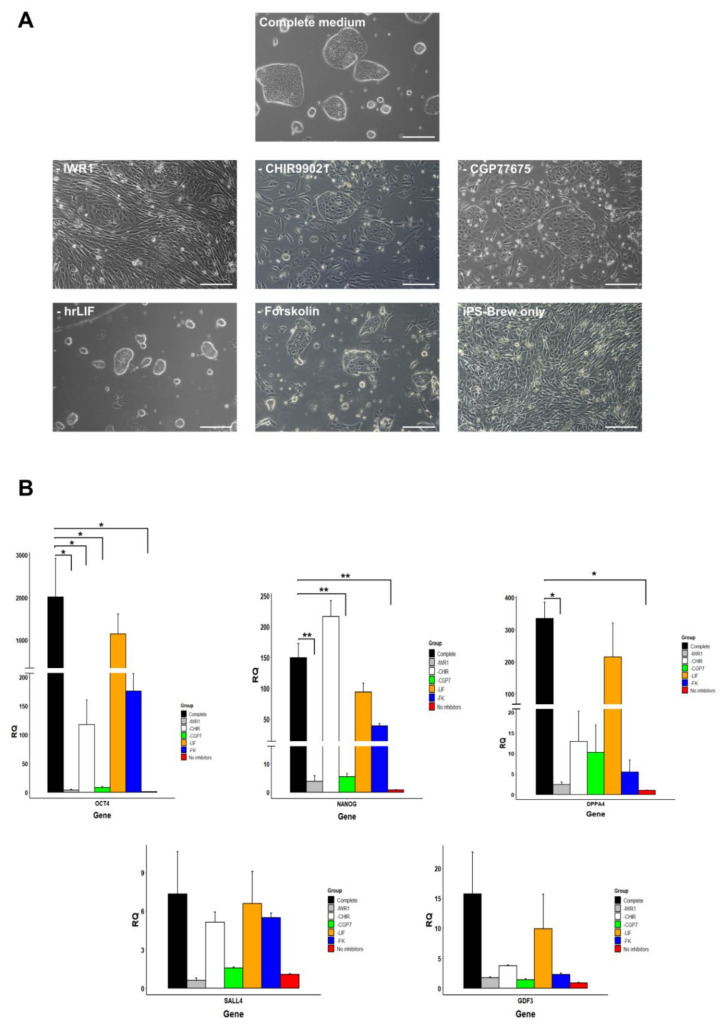
Role of small molecule inhibitors in maintaining pluripotency gene expression of marmoset iPSCs in long-term culture. (**A**) Marmoset iPSCs cultured either with full culture medium, with omission of individual factors (-IWR1, -CHIR99021, -CGP77675, -hrLIF, -Forskolin), or without any inhibitors and hrLIF. (All scale bars = 50 μm). (**B**) Pluripotency gene expression of marmoset iPSCs cultured without different inhibitors or hrLIF for 5–6 passages determined by relative quantitation qPCR. One of the lines cultured without any inhibitors and hrLIF was used as a reference. (Data are presented as mean + SEM. Statistically significant differences between experimental groups are indicated with asterisks as follows: * *p* < 0.05, ** *p* < 0.01).

**Figure 7 cells-09-02422-f007:**
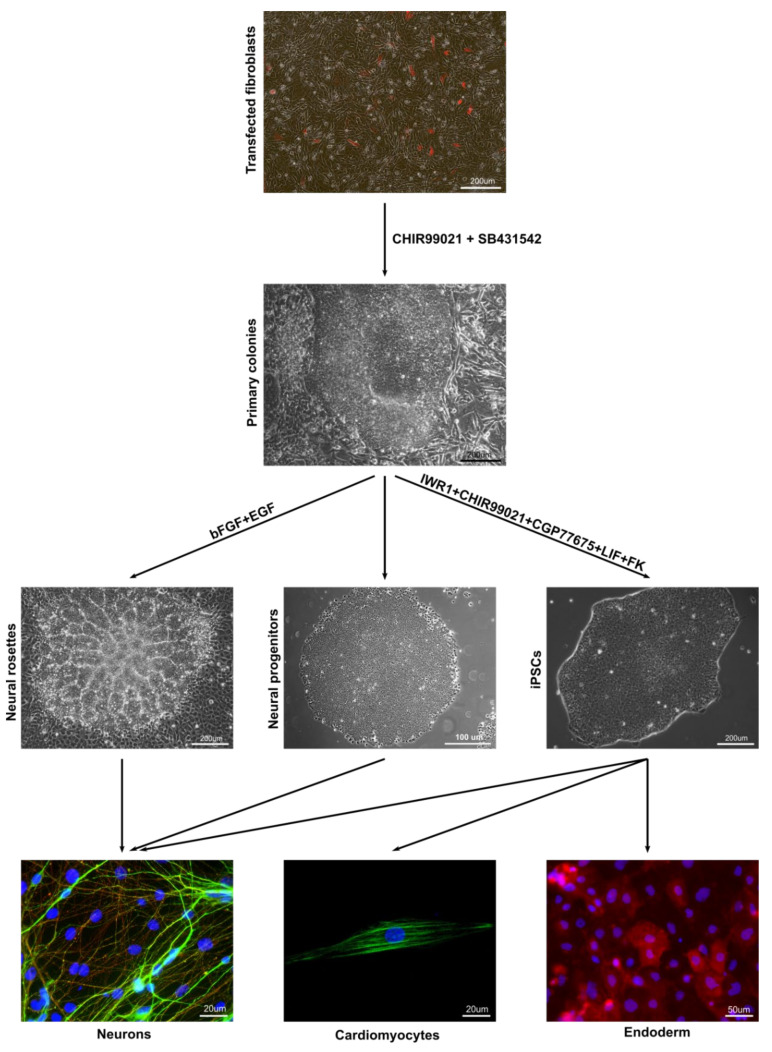
Paradigm for two-step flexible reprogramming of marmoset somatic cells to pluripotency or to the neural lineage. Transfection of somatic cells with vector carrying reprogramming transcription factors and culture in medium containing CHIR99021 and SB431542 leads to generation of intermediate primary colonies. These colonies can be maintained as neural progenitors with the same culture medium, directed into the neural lineage by culture with bFGF and EGF, or further reprogrammed to iPSCs by culture with IWR1, CHIR99021, CGP77675, leukemia inhibitory factor (LIF), and Forskolin.

**Table 1 cells-09-02422-t001:** Primer sequences.

Primers for RT-PCR			
Gene	Accession Code	Primer Sequences	Amplicon (bp)
OCT4A	ENSCJAT00000038869.2	Fw: GGC TTG GGG CGC CTT CCT TC	
		Rv: CAG GGT GAT CCT CTT CTG CTT C	503
NANOG	XM_002752302.3	Fw: GCC ACC TGA AGA TGT GTG AAG ATG AAT G	
		Rv: GGG TAG GCA TAA TGT AAA CAG AAC ACG	149
ZFP42	XM_003732392.3	Fw: GAAACCACGTCTGTGCAGAGTGTG	
		Rv: GCATGAGTTAGGATGTGGGCTTTCAG	256
GDF3	XM_002752299.3	Fw: GCTGGATGTAGCTAAGGATTGGAATG	
		Rv: GAAGTTAATGAATAGCTGGTGACGGTG	271
DPPA2	XM_008982501.2	Fw: CACTTTGCGGAACTGGTGTCAAG	
		Rv: CAGTCTTAGGCTGAACAGCTCTG	311
DPPA4	XM_002758824	Fw: GTCCATGGGAAAAGTCTCCCTGCAG	
		Rv: CACCACGGAATCCGACTCTCCAG	287
SALL4	XM_002747676	Fw: CTACTGACAGCGTTCCCAAACACCAG	
		Rv: CAGCATAGCAACAATCGTGATTGT	252
TERT	ENSCJAT00000074768.1	Fw: CTGCTCCTGCGTTTGGTGGATG	
		Rv: CGTCTGGAGGCTGTTCACCTG	405
SOX2	ENSCJAT00000016325.3	Fw: GCTCGCAGACCTACATGAACG	
		Rv: GCGGTCCGGCCCTCACAT	332
NESTIN	XM_008984607.2	Fw: GCGTTGGAACAGAGGTTGGAG	
		Rv: GTCTCAAGGGTAGCAGGCAAG	377
NOTCH1	XM_017963672.1	Fw: CTTACAGATGCAGCCACAGAACCTG	
		Rv: GCGGGCAATCTGAGACTGCATG	491
OTX2	ENSCJAT00000036430.2	Fw: GGGCTGAGTCTGACCACTTCG	
		Rv: GAGATGGCTGGTGACTGCATTG	657
GAPDH	XM_017976537.1	Fw: CACTGGCGTCTTCACCACCATG	
		Rv: GACACGGAAAGCCATGCCAGTG	305
**Primers for real-time qPCR**			
**Gene**	**Accession code**	**Primer sequences**	**Amplicon (bp)**
OCT4A	ENSCJAT00000038869.2	Fw: GGCGCCTTCCTTCCCCATGG	
		Rv: GGCGAGAAGGCAAAATCCGAAG	54
NANOG	XM_002752302.3	Fw: GCCACCTGAAGATGTGTGAAGATGAATG	
		Rv: GGATTCAGCCAGTGCTCAGAGTG	71
ZFP42	XM_003732392.3	Fw: GTGTCCCTTTGAAGGCTGTAGGAAG	
		Rv: GCATGAGTTAGGATGTGGGCTTTCAG	72
GDF3	XM_002752299.3	Fw: GCTGGATGTAGCTAAGGATTGGAATG	
		Rv:CAGTATCTCCAGGAATAGCCCGAAG	67
DPPA2	XM_008982501.2	Fw: CACTTTGCGGAACTGGTGTCAAG	
		Rv: CAGGCATATCTTGCTGTTGTTCAGG	113
DPPA4	XM_002758824	Fw: GTCCATGGGAAAAGTCTCCCTGCAG	
		Rv: GAACCCAGGCCTGACCAGCATG	76
SALL4	XM_002747676	Fw: CTACTGACAGCGTTCCCAAACACCAG	
		Rv:GCACGTTCTCCTTTAGCTTAGCTG	89
SOX2	ENSCJAT00000016325.3	Fw: GCTCGCAGACCTACATGAACG	
		Rv: GACTTGACCACCGAGCCCATG	103
NESTIN	XM_008984607.2	Fw: GCGTTGGAACAGAGGTTGGAG	
		Rv: GACATCTTGAGGTGCGCCAG	163
NOTCH1	XM_017963672.1	Fw: CATTCCAACGTCTCTGACTGGTCTG	
		Rv: GCGGGCAATCTGAGACTGCATG	77
OTX2	ENSCJAT00000036430.2	Fw: CCAACCATTGCCAGCAGCAGTG	
		Rv: GAAGAGGAGGTGGACAAGGGATCTG	89
GBX2	XM_017973818	Fw: GAGTAGCACCGCCTTCAGCATAG	
		Rv: GGGTAGCCGGTGTAGACGAAATGG	83
PAX3	XM_008999566	Fw: GCTGTCGGAGACCTCTTACCAG	
		Rv: GAACCGTGCTGCTGGGATCCG	68
MSX1	XM_003732530	Fw: GACGAACCGTAAGCCACGGACG	
		Rv: GCTCGGCGATGGACAGGTACTG	95
DLX5	XM_017975343	Fw: GCTCTCCTACCTCGGCTTCCTATG	
		Rv: GCCGTTCACGCCGTGATACTG	77
SOX10	XM_017963624	Fw: GACATCGGTGAGATCAGCCACGAG	
		Rv: CGGCAGGTACTGGTCCAACTCAG	80
SNAi1	XM_003732844	Fw: CTTTGCCGACCGCTCCAACCTG	
		Rv: GGAGCAGGGACATTCGGGAGAAG	107
18S	XM_002746395	Fw: ATTAAGGGTGTGGGCCGAAG	
		Rv: GAGTTCTCCTGCCCTCTTGG	81
**PCR primers for detection of the T7-VEE-OKS-iM-iTomato plasmid DNA**			
VEE_OKSiM_F1: CAC CAC TCT GGG CTC TCC CAT G			
VEE_OKSiM_R1: GTC CAG GTC CAG GAG ATC GTT G 388			

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
