# Peer review of "Controlling the Switch from Neurogenesis to Pluripotency during Marmoset Monkey Somatic Cell Reprogramming with Self-Replicating mRNAs and Small Molecules"

_cells, 2020, doi:10.3390/cells9112422_

Round 1
Reviewer 1 Report
The authors had already tried to do the response to the reviewer's comments as possible. The manuscript should be published.
Reviewer 2 Report
In this paper, Petkov et al describe successful generation of iPSCs from common marmoset monkeys on a two step reprogramming method. First, marmoset fibroblasts are converted to intermediate primary colonies by single transfection with VEE-mRNAs, and then primary colonies are converted to iPSCs by using customized media and feeder-free conditions. In addition, intermediate primary colonies have neural progenitor like properties and can be differentiated to neurons, astrocytes and oligodendrocytes as well as maintained as neural cells for long periods of time in vitro.
The manuscript presents a novel technology, and it is informative and well written. However, this reviewer has some concerns that need to be addressed before accepting it for publication at Cells:
The authors present a novel reprogramming technology of marmoset somatic cells that potentially alleviates some issues encountered with alternative technologies, such as genomic alterations and/or instability, persistence of episomal plasmid expression or increased stress due to sequential transfection steps. It would be also helpful for readers to discuss potential limitations of this novel approach, such as efficiency of the technology compared to other reprograming methods and additional caveats if present.
Why authors are using fetal fibroblasts? Have they tried reprogramming adult fibroblasts?
At the beginning of the MS it gets complicated to follow the whole design and cell types generated. It will be helpful to include a scheme at the beginning of the MS (in Figure 1, maybe combined with figure 1B) where all steps of the process and cell types obtained are indicated. Something similar to Figure 7 but at the beginning to help the reader following clearly all steps.
Figure 1D-F show images of intermediate primary colonies at DIV27 after transfection in which many non-reprogrammed fibroflasts are still present. Is there a way to eliminate those from your culture? They might affect the final fate of your colonies. Have you tried to FACsort the Tomato+ cells and replate them?
Figure 1G shows iPSC colony surrounded of many differentiated cells, probably primary intermediates or other differentiated cells. Have you characterized them? I wonder what is the efficiency of this second reprogramming step. Are many iPSCs obtained or most of the cells remain as primary intermediate colonies or other differentiated cells?
Figure 4A-B show neurons + for B-III-Tubulin and Nestin. Since Nestin also stains neural stem cells, can authors use a more specific marker such as MAP2 or NeuN?
Figure 5G-H show astrocytes that are GFAP+. Can authors include additional astroglial markers such as S100b, EAAT1 or AQP4?
Minor comment: legends on the graphs are too small and it is difficult to read them.
After considering these mentioned above comments, the MS would be suitable and appropriate for publication in Cells.
Reviewer 3 Report
The manuscript by Petkov, Behr et al. presents the generation of marmoset monkey iPSC lines using self-replicating mRNA vectors based on the Venezuelan equine encephalitis virus (VEE-mRNAs) and small molecules.
The authors claim a two-step reprogramming protocol first establishing intermediate primary colonies with neural progenitor-like properties and second their full conversion to iPSC colonies. The ‘rationale’ behind this was that (‘one-step’) protocols described for mRNA reprogramming of human cells failed to produce any marmoset iPSCs.
When I see it right from Figure 1B and the textual description the authors are cultivating from day 7/8 after the mRNA vector transfection until day 27/30 days in iPS-Brew medium with CHIR and SB small molecules to come to their intermediate primary colonies which have the neural lineage potential further evaluated in Figure 5. Further cultivation in iPS-Brew medium with a different small molecule supplementation then drives these colonies to full iPSC.
It is puzzling to me, why this setup is not discussed in the context of direct conversion of fibroblasts into expandable neural stem cells (e.g. Thier, Edenhofer et al., 2012 // Han, Schöler et al., 2012 Cell Stem Cell). Petkov, Behr basically claim the direct conversion of fibroblasts to neural progenitor cells by use of OSKM and iPS/CHIR/SB medium. This is somehow against the current understanding, because when using OSKM Thier, Edenhofer need the addition of instructive neural stem cell media conditions to pattern in this direction. So, changing from iPS to NSC medium makes the difference when initiating reprogramming with OSKM and then going down either towards the pluripotency or neural path. Here it is really striking, how a neural progenitor with tripotentiality as claimed in Figure 5 can be generated under ‘pure’ iPSC conditions. To elucidate this, I would recommend to change after day 7/8 in their protocol in NSC conditions (bFGF/EGF) like others and compare what kind of colonies are showing up at day 27/30 either under iPSC or NSC conditions. The results have to be discussed in the context of the direct neural reprogramming literature.
Besides this major issue the current Figures are appropriate to support the claims made. Figure 1 presents the protocol and the primary colony and iPSC morphology. Figure 2 present immunostainings and PCR analysis for pluripotency markers. Figure 3 presents RT-qPCR analysis of lineage-specific markers. Figure 4 presents the in vitro (EB) and in vivo (Teratoma) differentiation of the marmoset iPS cells. Figure 5 characterizes the neural tripotentiality of the primary colonies. Finally Figure 6 elaborates stability of iPSC long-term culture under different small molecule media combinations.
Global gene expression profiling of the primary colonies and final iPSC and comparison to neural stem/progenitor cells and iPSC data would increase the impact of the manuscript significantly, however we are aware of the issue that not every lab would like to dig into RNAseq for a ‘simple’ iPSC generation paper.
The literature concerning iPSC generation with small molecules is abundantly represented, the whole aspect of direct reprogramming to NSC is missing as discussed above. Please revise your manuscript.
This manuscript is a resubmission of an earlier submission. The following is a list of the peer review reports and author responses from that submission.
Round 1
Reviewer 1 Report
cells-830822
“Controlling the switch from neurogenesis to pluripotency during marmoset monkey somatic cell reprogramming with self-replicating mRNAs and small molecules”
The authors reported that iPSCs derived from common marmoset monkey fibroblasts using self-replicating mRNA vectors, especially based on a two-step strategy. In the first step, the authors induced a neural progenitor-like state (intermediate primary colony) with chir99021 and sb431542. In the second step, they converted these colonies into pluripotent stem cells with customized iPSC medium. While the point of views of authors for neural lineage mediated reprogramming are interesting, the several important points are neglected of this manuscript raises serious concerns:
1) Insufficient description for research background
The authors mentioned that they used chir99021 and sb431542 because they failed to obtain any iPSC lines with conventional iPSC medium. These results are acceptable because the reprogramming efficiency of iPSC is very low without small molecules supplement. But I think the authors needed to explain the reasons why they select chir99021 and sb431542 based on the previous reports and their study. And many reports showed the reprogramming efficiency for comparison of induction conditions. Furthermore, additional comments are also needed to understand why neural lineage intermediates are induced in this condition. As much as I know, several research groups have been reported that chir99021 and sb431542 not only enhance reprogramming processes but also support the self-renewal of neural stem cells. In addition, authors need to review the two-step reprogramming strategy in the discussion part. As much as I know, several reports showed that two-step reprogramming strategy is useful for higher reprogramming efficiency. (J Mol Cell Biol . 2012 Feb;4(1):59-62. Cell Res. 2019 Sep;29(9):696-710. )
Authors also need to comment on the reasons IWR1, CHIR99021, CGP77675, rhLIF, and forskolin are selected for customized marmoset iPSC medium. Based on my knowledge, this combination of small molecules is very similar to expanded potential stem cells culture condition which composes with gsk-3/Wnt/Src inhibitors with LIF (Nat Cell Biol. 2019 Jun;21(6):687-699.). I think authors needed to comment about their customized marmoset iPSC medium.
2) Authors also elucidate the effects of growth factors.
Authors need to show the function of each small molecules for self-renewal and reprogramming for iPSCs. Withdrawal of one molecule from customized marmoset iPSC medium and analyze the gene expression profiles might explain the function of each small molecules.
3) insufficient comment for lineage-specific marker expression.
In figure 3A, SOX2, OTX2, and GBX2 are upregulated in iPSCs compared with both cjFFs and primary colonies. As much as I know, these genes are related to neural patterning of brains. I could not understand the meaning of these genes upregulation in pluripotent stem cells. Considering that these genes are lineage-specific genes, but not pluripotent related genes, this gene expression pattern should be explained.
4) Insufficient analysis of intermediate primary colonies
As shown in figure 5c, intermediate primary colonies derived cells are SOX2 and NESTIN positive, which is one of the characteristic s of neural stem cells (NSCs). These cells could differentiate in β-III-Tubulin positive neurons. In this position, the authors need to conclude the identity of intermediate primary colonies. The authors did not show astrocyte and oligodendrocyte differentiation potentials of, intermediate primary colonies. If astrocyte and oligodendrocyte differentiation are impossible, these cells are neuronal progenitors. If astrocyte and oligodendrocyte differentiation are possible, these cells could possibly NSCs. If authors wanted to insist on these cells and NSC-like cells, authors have to show tri-potent differentiation capacity, self-renewal ability, and neural-specific differentiation, but not endoderm and mesodermal differentiation.
5) There is no comment about the Institutional Review Board (IRB) or research approval of marmoset monkey cells.
Reviewer 2 Report
This is a very interesting study concerning the derivation of iPSCs from marmoset fibroblasts with VEE-mRNAs carrying the human OCT4, KLF4, SOX2, and c-MYC and cultured in the presence of small molecule inhibitors CHIR99021 and SB431542. The iPSCs had the potential directed towards the neural lineage or further reprogrammed to pluripotency. The results provide the platform for the animal disease model for preclinical testing of iPSC-based therapies and also establish the in vitro system for the signal transduction pathways in primates. The manuscript was well written and the sound experiment design and solid result were also presented. These data should be published.